# Partnering with Aboriginal and Torres Strait Islander Peoples: An Evaluation Study Protocol to Strengthen a Comprehensive Multi-Scale Evaluation Framework for Participatory Systems Modelling through Indigenous Paradigms and Methodologies

**DOI:** 10.3390/ijerph20010053

**Published:** 2022-12-21

**Authors:** Grace Yeeun Lee, Julie Robotham, Yun Ju C. Song, Jo-An Occhipinti, Jakelin Troy, Tanja Hirvonen, Dakota Feirer, Olivia Iannelli, Victoria Loblay, Louise Freebairn, Rama Agung-Igusti, Ee Pin Chang, Pat Dudgeon, Ian Bernard Hickie

**Affiliations:** 1Brain and Mind Centre, Faculty of Medicine and Health, The University of Sydney, Sydney, NSW 2050, Australia; 2Centre of Best Practice in Aboriginal and Torres Strait Islander Suicide Prevention, School of Indigenous Studies, University of Western Australia, Perth, WA 6009, Australia; 3Computer Simulation & Advanced Research Technologies (CSART), Sydney, NSW 2021, Australia; 4College of Medicine and Public Health, Flinders University, Adelaide, SA 5042, Australia; 5National Centre for Epidemiology and Population Health, The Australian National University, Canberra, ACT 2601, Australia

**Keywords:** social and emotional wellbeing, Aboriginal and Torres Strait Islander, Aboriginal participatory action research, Indigenous research and evaluation methodologies, youth mental health, monitoring and evaluation, participatory systems modelling, systems modelling and simulation, stakeholder-based modelling, community empowerment research

## Abstract

The social and emotional wellbeing of young Aboriginal and Torres Strait Islander peoples should be supported through an Indigenous-led and community empowering approach. Applying systems thinking via participatory approaches is aligned with Aboriginal and Torres Strait Islander research paradigms and can be an effective method to deliver a decision support tool for mental health systems planning for Indigenous communities. Evaluations are necessary to understand the effectiveness and value of such methods, uncover protective and healing factors of social and emotional wellbeing, as well as to promote Aboriginal and Torres Strait Islander self-determination over allocation of funding and resources. This paper presents modifications to a published evaluation protocol for participatory systems modelling to align with critical Aboriginal and Torres Strait Islander guidelines and recommendations to support the social and emotional wellbeing of young people. This paper also presents a culturally relevant participatory systems modelling evaluation framework. Recognizing the reciprocity, strengths, and expertise Aboriginal and Torres Strait Islander methodologies can offer to broader research and evaluation practices, the amended framework presented in this paper facilitates empowering evaluation practices that should be adopted when working with Aboriginal and Torres Strait Islander peoples as well as when working with other diverse, non-Indigenous communities.

## 1. Introduction

We acknowledge the longest continuing and strongest standing culture by paying deepest respect to the sovereign owners of Australia. We acknowledge that the research presented through this paper is only possible because of the ancient wisdom, expertise, and strengths of Aboriginal and Torres Strait Islander Elders, communities, cultures, research and ways of knowing, and custodianship to the lands, waters, and skies. 

We acknowledge that Aboriginal and Torres Strait Islander peoples have not always been respected and recognized in academia; obstructed from opportunities within its institutions due to colonization and racism. Thus, all authors listed on this paper are committed to enabling and nurturing Aboriginal and Torres Strait Islander self-determination, leadership, and empowerment of Indigenous knowledges in the broader structures of the academy. Though not all authors are of Aboriginal and Torres Strait Islander descent, we recognize that each of the authors bring different strengths and expertise. Aboriginal and Torres Strait Islander knowledge has taken precedence throughout the paper, affirmed by the expertise of the Aboriginal and Torres Strait Islander partners who are also authors of this paper (JT, TH, DF, PD). 

*Aboriginal and Torres Strait Islander peoples* is utilized throughout the paper to refer to the First Peoples of Australia. However, *First Nations* and *Indigenous* are also utilized throughout this paper when reflecting an international context. 

### 1.1. Social and Emotional Wellbeing of Young Aboriginal and Torres Strait Islander Peoples 

The concept of social and emotional wellbeing is deeply rooted in Indigenous cultures [1], and holistically encompasses the broader social, historical, political, and cultural determinants of Aboriginal and Torres Strait Islander mental, spiritual, and physical health [2,3,4]. Aboriginal and Torres Strait Islander peoples have endured and continue to resist colonization and cultural genocide [5], including dispossession and removal from their culture and Countries [1]. Challenges that stem from complex “colonial, political, social, and economic histories” [5] have resulted in ongoing adverse consequences to the social and emotional wellbeing of Aboriginal and Torres Strait Islander men, women, young people, and children [1,2].

Young Aboriginal and Torres Strait Islander peoples are especially vulnerable, and “a new destructive phenomena” is observed in the loss of young Indigenous Australians to suicide [6]. Mental ill health was identified as the most important health issue for young Australians in 2014 [7], described as an international crisis in 2018 [8], and globally exacerbated in recent years as a result of extreme weather events, dire economic trends, and prolonged impacts of COVID-19 on the social and emotional wellbeing of young people [9,10,11,12]. It is also recognized that Aboriginal and Torres Strait Islander adolescents and adults experience higher levels of health and social disparities [13]. For example, three times as many young Aboriginal and Torres Strait Islander peoples under 18, and 12 times as many Aboriginal and Torres Strait Islander peoples under 15 years of age die by suicide compared to non-Indigenous Australians [14], with actual rates believed to be significantly higher than reported rates [15]. In addition, 53% of Indigenous peoples reported living with financial stress in 2018–2019 (i.e., unable to raise $2,000 AUD in one week for emergency situations), an increase from 48% in 2014–2015 [16]. In 2020, 29% of adult prisons were comprised of Aboriginal and Torres Strait Islander peoples (despite representing only 3% of the total Australian population), and young Aboriginal and Torres Strait Islander peoples made up 48% of the youth prison population (despite representing 6% of the total Australian population) [17]. A 2021 report of the largest national annual survey in Australia (20,207 young people aged 15–19 years old) reported that approximately half of Aboriginal and Torres Strait Islander respondents (~47% compared with 34% of non-Indigenous respondents) reported experiences of unfair treatment, with more than half (~53%) attributing their race/cultural background to their negative experiences [18].

Adopting a multifaceted approach to Aboriginal and Torres Strait Islander social and wellbeing that takes into consideration the “body, mind and emotions, family and kinship, community, culture, Country, and spirituality” is necessary to have significant impact and strengthen individual, family, and community wellbeing [19]. As such, there is a need for all health care providers, administrators, and policy makers to better understand and address drivers behind such disturbing gaps in outcomes in young Aboriginal and Torres Strait Islander peoples compared to non-Indigenous Australians.

There is a welcomed and necessary shift in narrative against “problematic constructions of Indigenous peoples as the sole cause of their poor [health] status” [5], and there is growing acknowledgement of how cultural genocide, racism, and colonization have resulted in failure to recognize Indigenous values, beliefs, and laws in the broader health and social systems [2,20]. However, there is an additional need to recognize the everyday impacts colonization continues to have on Aboriginal and Torres Strait Islander peoples’ life opportunities, wellbeing, healing, self-determination, and ongoing lived experience. For instance, recognizing the everyday impacts of colonization sheds further insight on the recurring marginalization experienced by Aboriginal and Torres Strait Islander peoples in accessing high quality (e.g., culturally acceptable, Indigenous-led, etc.) primary health care [21,22], which is oftentimes the first point of entry to receiving social and emotional wellbeing support [23]. The unavailability for Aboriginal and Torres Strait Islander peoples to access high quality primary health care services is in essence, failing to meet international human rights declarations to access the highest attainable standard of health and wellbeing [24,25,26,27].

Aboriginal Community Controlled Health Services (ACCHS) and other Aboriginal and Torres Strait Islander organizations are available throughout Australia to provide various culturally safe primary health care services for local Indigenous communities and support better outcomes in Aboriginal and Torres Strait Islander peoples [4,22]. However, there are still missed opportunities as youth services, mental health or social and emotional wellbeing services, and alcohol, tobacco and other drug services are reported as the main service gaps by Indigenous primary health care organizations [28]. There are also gaps where Aboriginal and Torres Strait Islander peoples may not be able to access Aboriginal health services [28,29,30], and mainstream Australian youth mental health services are failing First Australians as these services reflect colonized beliefs and standards [31,32]. The unavailability of culturally appropriate health services can further exclude Aboriginal and Torres Strait Islander peoples from accessing care when required [32], as mental health and psychological distress in Western medicine is generally focused on symptomology, and rarely considers broader cultural and spiritual elements to social and emotional wellbeing as is emphasized in Indigenous ideology. There are direct impacts attributed to culturally unsafe interactions and understandings in health settings including lower levels of health investigations, interventions, and medical prescriptions adjusted for need, and there is an ongoing anticipation of poor health outcomes and mistrust when accessing mainstream services [33]. As such, 30% of Aboriginal and Torres Strait Islander peoples reported between 2018–19 that they did not seek a health care provider in the past 12 months despite requiring support, with 46,180 Indigenous Australians not seeking care from a counsellor when needed [28].

### 1.2. Participatory Systems Modelling (PSM) to Support the Social and Emotional Wellbeing of Young Aboriginal and Torres Strait Islander Peoples

Significant structural changes to the broader health and social systems are required to support the social and emotional wellbeing of Aboriginal and Torres Strait Islander young peoples. Systems modelling and simulation are increasingly being used in mainstream mental health services planning to understand how best to invest limited resources to deliver the greatest impacts on health, social, and economic outcomes [34]. Such decision support tools offer significant promise for informing investments to strengthen the mental health system (including broader social and cultural contexts) to ultimately improve the social and emotional wellbeing of young Aboriginal and Torres Strait Islander peoples. However, the development of these decision support tools requires an Indigenous-led and community empowering approach, enabling positive growth and self-determination [34,35,36]. Systems modelling via a participatory approach aligns with Aboriginal and Torres Strait Islander research paradigms such as the emphasis on building active research partnerships with community members [37], as well as focusing more holistically on the collective that make up the system, rather than on individual components. However, these methods have yet to be implemented to support the social and emotional wellbeing of young Aboriginal and Torres Strait Islander peoples, operating under Indigenous governance and leadership.

A team of multidisciplinary researchers at The University of Sydney’s Brain and Mind Centre are embarking on the *Right care, first time, where you live* research program (hereafter referred to as ‘Program’). This five-year national youth mental health Program aims to enhance the social and emotional wellbeing of young people across Australia, with a focus on supporting evidence-based policy and funding decision making to improve broad youth mental health outcomes including youth engagement in education and employment [38]. The Program also places an emphasis on empowering local communities, equipping local stakeholders with the tools, resources, and training opportunities to support more informed, transparent, and inclusive approaches to decision making whilst taking into consideration broader social, historical, political, and cultural determinants of mental health [39].

This will be achieved through the application of systems modelling and simulation methods through a participatory approach (more commonly referred to as participatory systems modelling or PSM) in eight geographically diverse communities across Australia. To ensure Program sites best represent the diversity in Australia, two sites will focus on young Aboriginal and Torres Strait Islander peoples, creating opportunities to partner with local Indigenous communities in an urban as well as in a rural region. Through the PSM process, eight local system dynamics models will be co-developed. System dynamics modelling is the most appropriate method to apply for the Program as these models can provide a necessary predictive planning framework at the population health level for local communities, which is typically underutilized in mental health systems planning [40]. These models can simulate hypothetical ‘what-if’ scenarios of likely health, social, and economic impacts of individual (e.g., ACCHS) and/or a combination of programs and services (e.g., ACCHS combined with community-based acute care services). Importantly, the PSM process focuses on working with local community stakeholders as co-researchers, empowering communities to prioritize what should be incorporated into the system dynamics models, such as which programs and services should be included within the model. Supporting local community members to prioritize what should be included in the system dynamics models ensures that the final tool, which can simulate various ‘what-if’ scenarios of likely health, social, and economics impacts, reflects the priorities of the community. In addition to the development of the system dynamics models, participatory processes are embedded throughout the Program’s participatory modelling [41], evaluation [42,43], and economic analysis [44] processes to achieve genuine community inclusion in all phases of the Program, ensuring Aboriginal and Torres Strait Islander communities are empowered and their priorities appropriately reflected [37].

The selection of sites participating in the Program is detailed elsewhere [37]. To summarize, a rigorous selection process was implemented as part of the Program’s four-phase stakeholder engagement framework, where expressions of interests were sought from primary health organizations across Australia (i.e., Primary Health Networks). A site selection matrix was used which included criteria to ensure a diversity of sites based on geographic locations [37]. Primary health organization sites with strong Indigenous governance structures and an active commitment to implementing their Reconciliation Action Plan were selected to ensure that existing Indigenous leadership and governance can drive the process of embedding locally relevant Aboriginal and Torres Strait Islander community perspectives and needs throughout the Program [37]. This Program is the first to partner with Aboriginal and Torres Strait Islander communities to co-develop a system dynamics model that best reflects the strengths and complexities of the broader health and social systems that impact young Indigenous peoples, through a participatory process of empowering and decolonizing mental health services, workforces, and systems.

### 1.3. Maximizing Opportunities in PSM—Culturally Appropriate and Empowering Evaluation Approaches by Partnering with Aboriginal and Torres Strait Islander Communities

Evaluations have been described as the “quiet movement to make government fail less often” [45]. Despite the importance of evaluations, programs that impact Aboriginal and Torres Strait Islander peoples have been inappropriately evaluated—with Indigenous ways of knowledge, practice and culture excluded—which has limited improvements to their health within Australia and in other colonized First Nations populations [46]. Evaluations are also often inadequately planned with negligible input from Aboriginal and Torres Strait Islander peoples themselves, which affect their quality and usefulness [47]. When evaluations are conducted, findings are seldom published which contributes to an ongoing cycle of policy makers reporting a lack of evidence and Aboriginal and Torres Strait Islander peoples reporting the burden of being over-researched in a culturally inappropriate way [26,48,49].

When conducted poorly, evaluations lead to wasted resources, ongoing implementation of potentially harmful policies and programs, and growing mistrust from Indigenous peoples in the value of research and evaluation [46,50,51]. On the other hand, rigorous evaluations play a critical role in the ongoing investments on Indigenous-led and community empowering programs, strengthening transparency of the evidence base to support improvements in policies and programs designed to benefit Aboriginal and Torres Strait Islander peoples [50]. Evaluations can also increase accountability of policy makers and funding agencies to ensure Indigenous programs and services are sufficiently resourced, as well as to identify and mitigate unintended negative consequences of harmful policies, programs and practices [26].

Through the advocacy of Aboriginal and Torres Strait Islander leaders and communities, Australia’s Commonwealth Government has recognized that government policies and programs designed to improve the lives of Indigenous peoples are “not working as well as they need to” [47]. This has prompted increased investments over the past decade to strengthen the monitoring, evaluation, and dissemination of programs targeted at improving Aboriginal and Torres Strait Islander health and wellbeing [26]. This has since been followed by the 2020 Indigenous Evaluation Strategy [26,48]. This Strategy, along with a practical implementation guide, has been published as a result of extensive consultations through the Australian Productivity Commission. The consultations, led by Commissioner Romlie Mokak, a Djugan man and member of the Yawuru people [52], revealed a set of principles and toolkits for appropriate evaluation, and emphasizes the necessity for genuine engagement and partnership with Aboriginal and Torres Strait Islander peoples [47,48,53].

Just as there has been renewed attention to improving evaluation efforts in Aboriginal health research, the same investment is required in PSM evaluations which are generally either poorly conducted (e.g., lacking detail on transparent methodological approaches) or disregarded entirely [54,55]. To address this gap in PSM research, a comprehensive multi-scale evaluation framework has been developed [43] and translated into a study protocol for the Program [42]. In summary, the Program’s evaluation seeks to longitudinally understand the (i) feasibility, (ii) value, (iii) change & action (impact), and (iv) sustainability of PSM processes, with participatory action research (PAR) principles embedded to support improvements of the PSM process through a more equitable manner [42,43].

Though the development of the current evaluation framework is a positive attempt to improve PSM evaluation efforts [43], there is opportunity to strengthen the Program’s evaluation approach by reflecting Indigenous perspectives, priorities, and expertise when undertaking PSM with Aboriginal and Torres Strait Islander peoples. The strengthened evaluation approach can also be applied to the Program when working with other non-Indigenous communities. The work presented in this paper will maximize the opportunities and generate insights for future social and emotional wellbeing research both by the Program team and to an international field of PSM research. This paper will also contribute to a growing field of implementation research to support culturally appropriate and thoughtfully designed evaluations of Aboriginal and Torres Strait Islander policies and programs [48], which encourages full participation of Indigenous communities, organizations, and broader health and wellbeing sectors [26].

### 1.4. Aims and Objectives

There are two key aims of this paper. Firstly, this paper aims to apply knowledge from landmark Aboriginal and Torres Strait Islander frameworks, guidelines, and recommendations, such as the 2020 Indigenous Evaluation Strategy [48], in the context of a PSM evaluation study protocol for our Program. Specifically, we aim to describe efforts to develop a culturally appropriate evaluation process that amplifies opportunities to include ongoing Aboriginal and Torres Strait Islander input, expertise, and perspectives. An additional aim of this paper is to test the cultural appropriateness of the current PSM evaluation framework developed by the first author (GYL) [43], and present a modified framework that is culturally relevant to adopt in PSM programs that include Aboriginal and Torres Strait Islander peoples.

A broad evaluation approach is described to support the replication of methodologies in future PSM programs undertaken with both Aboriginal and Torres Strait Islander peoples as well as with other non-Indigenous communities. In this way, we acknowledge that learning to modify evaluation methodologies through working with Indigenous communities may hold lessons for PSM modelling processes more broadly. Importantly, the presented evaluation process and framework is designed to be flexibly modified to invite Aboriginal and Torres Strait Islander communities to have complete ownership over the PSM evaluation process. Thus, Aboriginal PAR (APAR) principles are reflected in every element of the proposed study to empower Indigenous communities, including young people, through a strengths-based approach [56,57].

## 2. Materials and Methods

This study has been approved by the Sydney Local Health District Human Research Ethics Committee (Protocol No X21-0151 & 2021/ETH00553) on 5 July 2021, and by the Aboriginal Health & Medical Research Council of NSW (1875/21) on 9 February 2022. The described evaluation study has also been approved by and reflects the inclusion, lived priorities, and expertise of our Aboriginal and Torres Strait Islander partners, who are also authors of this paper (JT, TH, DF, PD).

The Program’s current PSM evaluation study protocol is described elsewhere [42]. In summary, a comprehensive multi-scale evaluation framework is applied drawing on PAR principles to longitudinally understand the (i) feasibility, (ii) value, (iii) impact, and (iv) sustainability of the PSM process (which involves a series of workshops with each of the participating eight communities to co-develop eight local youth mental health system dynamics models) [42]. As two of these models will reflect the priorities and needs of local Aboriginal and Torres Strait Islander communities, our currently published evaluation study protocol will be modified when working with these communities. The following sections will describe modifications to the currently published evaluation protocol, ensuring a culturally appropriate evaluation process in the Program’s Indigenous focused sites as well as in the other participating sites.

### 2.1. Study Design and Setting

The Program’s currently published evaluation study protocol has alignment to Aboriginal research paradigms and methodologies, with PAR principles embedded to drive reflexivity and action, enabling continuous improvement of the PSM process [37,42]. When working with Aboriginal and Torres Strait Islander communities, the PSM evaluation approach will be modified to ensure that APAR methodologies are reflected. APAR is informed by Indigenous Standpoint Theory and is “designed to centre and increase Indigenous voice and ‘epistemic self-determination’ in Indigenous research and psychology” [56]. Thus, APAR methodologies incorporated into the modified evaluation process promotes Aboriginal and Torres Strait Islander peoples’ self-determination by ensuring opportunities are created to empower participating Indigenous peoples, leaders, and communities to speak for and of themselves [58], encompassing Indigenous beliefs, ethics, paradigms, and methodologies. For instance, APAR allows Aboriginal and Torres Strait Islander peoples participating in the Program to nominate evaluation points that are most meaningful for their communities, rather than researchers imposing their own interests, research questions, and methods. This also enables genuine partnerships with Aboriginal and Torres Strait Islander communities as co-researchers throughout the PSM process, from the design, analysis, through to dissemination of learnings [56,59,60], as well as enabling the Program to reflect the local priorities of Aboriginal and Torres Strait Islander communities rather than those of the Program research team [60]. Opportunities for research and evaluation capacity building will be encouraged, aligning with principles of reciprocity, and to acknowledge the rights of Aboriginal and Torres Strait Islanders to self-determination in the achievement of research [60].

### 2.2. Community Inclusion & Recruitment Procedure

Our published evaluation study protocol describes how up to 55 participants will be included per participating site of the Program to capture diverse perspectives [42]. An inclusion criterion is also described, which details considerations for inclusion such as age (i.e., ≥14 years) and language (i.e., English proficiency) [42]. When working with Aboriginal and Torres Strait Islander communities, the evaluation process will be modified so that researchers are guided by local knowledge and expertise. This not only empowers local communities to provide input, but it also supports flexibility in the evaluation process as every participating local community will have varying priorities and needs.

In addition to working in collaboration with the primary partner organization as described in the currently published Program evaluation protocol [42], there will be a focus on leveraging Indigenous leadership and governance such as the establishment of local Indigenous community reference groups [56]. This approach recognizes the “centrality of Indigenous self-determination and leadership” [60]. Active engagement with Aboriginal and Torres Strait Islander leaders at the earliest stage possible is also imperative. Thus, engagement with key Indigenous stakeholders will commence during the early evaluation design and planning process, and effective engagement strategies will be applied to amplify and respect the strengths and capacities of local Indigenous leadership [37].

Program researchers will be guided by local Indigenous leadership and perspectives, through a local Aboriginal and Torres Strait Islander reference group (or equivalent) on appropriate recruitment and consenting procedures. This is to ensure validity in the Program PSM evaluation recruitment and consenting procedures from both an individual participant as well as the community level [61]. The Program team will additionally seek formal cultural guidance, training, immersion, and/or supervision from local Aboriginal and Torres Strait Islander leaders when possible prior to commencing evaluation, to ensure that researchers engage with each community with respect and acknowledgement of existing cultural strengths and expertise. A formal research agreement has also been executed with a national Aboriginal and Torres Strait Islander suicide prevention research centre (led by PD) which enables personnel support to ensure best research practices are adopted in the Program when engaging and working with Aboriginal and Torres Strait Islander communities.

### 2.3. Ownership and Control over Indigenous Data (Data Sovereignty)

Ownership of data (data sovereignty) is a longstanding concern in Aboriginal and Torres Strait Islander research [61], and the complexities of data sovereignty are recognized [62]. At the earliest stage of engagement possible, yarning and consultations with local Indigenous leadership and governance will be facilitated to acknowledge their authority over the data and discuss management, use, and dissemination of research data whilst still acknowledging respect for confidentiality. This will not only increase transparency of the Program evaluation process, but it also allows opportunities to openly ask critical questions related to data management. This is particularly significant as participating Aboriginal and Torres Strait Islander peoples are co-researchers and should have the ability to make joint decisions regarding the collection, access, analysis, and dissemination of research data [61].

Sufficient time will also be allocated to work with Indigenous leadership and governance to ensure respectful and mutual understanding is reached regarding data sovereignty, favoring an approach that empowers local Indigenous communities and strengthens research outcomes. As such, data sovereignty has been included in the Program’s written research collaboration agreements with the sites to allow for respectful negotiations to clarify and secure rights in data.

### 2.4. Data Collection Process

A mixed methods approach is described in the current Program evaluation protocol published by the first author (GYL), which includes collection of data through gamified online surveys, semi-structured interviews, and other qualitative data including research observations and recordings from the PSM workshops, meetings, reflections, and field notes [42]. Though some of these methods such as surveys are described as useful data collection methods in the 2020 Indigenous Evaluation Strategy, additional forms of data collection that magnifies Aboriginal and Torres Strait Islander paradigms are acknowledged. These include both qualitative methodologies such as dadirri (listening), ganma (knowledge sharing) [53], and yarning (sharing stories that “respect and honor in a culturally safe environment”) [63,64] as well as quantitative methodologies such as nayri kati (generating good statistical data through an Indigenous lens) [64]. Though non-Indigenous researchers can use Aboriginal and Torres Strait Islander research techniques such as yarning as an interview technique, this does not mean that they can engage or lead all Indigenous methodologies [65]. Thus, the Program team will work under guidance from local Aboriginal and Torres Strait Islander leaders as well as other Indigenous academics and experts (JT, TH, DF, PD) on which techniques are appropriate to implement by whom (e.g., co-led by an Aboriginal and/or Torres Strait Islander researcher from the site), with an evidence-based practice (learning from published work) and a practice-based evidence (learning from published and unpublished practical procedures) approach favored to continuously refine and improve the Program’s data collection processes [66].

Additionally, though the current data collection tools for the Program’s PSM evaluation have been extensively designed and tested with diverse participants, including the Brain and Mind Centre’s Youth Lived Experience Working Group (which includes representation of an Aboriginal young person) [42,67], it is acknowledged that these tools may not be appropriate to implement in local Indigenous communities. Therefore, expertise from local Indigenous leadership and governance will guide whether the evaluation tools are appropriate, what changes need to be made, as well as the process for testing and implementing the data collection methods. Full versions of the Program PSM evaluation tools, such as survey and semi-structured interview questions are available as Supplementary Material in our currently published evaluation study protocol [42].

### 2.5. Data Analysis Plan

The emergence of strong Indigenous research and evaluation paradigms and methodologies have primarily focused on data collection [68]. There is recognition that data analysis requires attention to ensure that all research phases go through a process of decolonization [68]. The Australian Institute of Aboriginal and Torres Strait Islander Studies’ (AIATSIS) Code of Ethics for Aboriginal and Torres Strait Islander Research (2020) recommends meetings with project partners and participants to discuss research results and analysis, creating opportunities for stakeholders to challenge the analysis and to provide additional perspectives [69]. Thus, the Program team will be guided by Indigenous leadership and governance, such as local Indigenous community reference groups, to ensure full participation of communities, critical inquiry, and strength-based approaches are regarded in the collection, access, analysis, and dissemination of research data [56,59,60,61]. This method of working with Aboriginal and Torres Strait Islander communities as co-researchers in the analysis of research data can also support appropriate data management practices—particularly data sovereignty and control—enabling Indigenous self-determination [61]. Self-determination includes the ability to collect and analyze data, and pursue research questions “that meets the needs of the Indigenous communities themselves, rather than the desires, ambitions and liberal-thinking non-Indigenous researchers” [61].

### 2.6. Data Security and Protection

There are growing concerns in Aboriginal and Torres Strait Islander research around the safety (e.g., security, confidentiality, privacy, etc.) of data as cloud-based storage is becoming more widely adopted [70]. Thus, the progressive development of data infrastructure and security systems is encouraged [70]. As such, the evaluation study has approval by the Sydney Local Health District Human Research Ethics Committee and by the Aboriginal Health & Medical Research Council of NSW to store data in accordance with The University of Sydney’s data management procedures, which includes the storage of data in the Research Data Store provided by The University of Sydney. This system is regularly backed up, has built in redundancy, complies with information security standards, and is supported by the University’s Research Data Management Policy [71] and Research Data Management Procedures [72].

## 3. Discussion

The Australian Government has committed to improving the social and emotional wellbeing of Aboriginal and Torres Strait Islander peoples through initiatives such as the National Strategic Framework for Aboriginal and Torres Strait Islander Peoples’ Mental Health and Social and Emotional Wellbeing 2017–2023 [73]; the 2020 Mental Health Productivity Commission Inquiry Report [74]; the national Closing the Gap strategy including the 2020 national agreement signed by the National Federation Reform Council and the Coalition of Peaks which will target social and emotional wellbeing as one of the five policy priority areas [75], and; most recently, the National Partnership Agreement for Mental Health and Suicide Prevention signed in March 2022 by the Commonwealth, State, and Territory governments to work in partnership to improve mental health for all Australians and reduce the rate of suicide toward zero [76]. The development of local system dynamics models through the Program’s participatory process described can support the social and emotional wellbeing of young Aboriginal and Torres Strait Islander peoples by equipping local leaders with an interactive predictive planning tool to support informed strategic decision making. Knowledge acquired from the eight regional system dynamics models will also enable enhanced understanding, communication, and assistance to national leaders through the deployment of a national model to best support social and emotional wellbeing strategies. Evaluations throughout the PSM process play an important role in understanding the effectiveness of such programs to uncover protective and healing factors that support social and emotional wellbeing, and importantly, to ensure that funding and resourcing is prioritized appropriately.

Research and evaluation protocols are typically slightly adjusted (or not adjusted at all) when working with Aboriginal and Torres Strait Islander communities leading to dire consequences. We challenge this practice by going through a process of decolonization as researchers [77], supported by APAR to enable critically reflexive praxis, ensuring our Program PSM evaluation nurtures Aboriginal and Torres Strait Islander self-determination, leadership, and knowledges. The methods in which the Program team will engage in a process of decolonization will be supported by Aboriginal and Torres Strait Islander expert guidance, and may include collective yarning circles and reflexive journal entries [77]. To ensure that local Indigenous peoples, communities, perspectives, priorities, and expertise are acknowledged and empowered [48,49], this paper presents modifications to our Program’s current evaluation approach by reflecting Aboriginal and Torres Strait Islander research and evaluation methodologies and paradigms. This enabled changes to the Program’s previous evaluation study design, strengthened through APAR methodologies to support an “empowering, developmental and transformative strategy” [56]. We argue that the modified evaluation approach presented in this paper strengthens the field of PSM evaluation more broadly, which has applications not only when PSM evaluations are undertaken with Aboriginal and Torres Strait Islander communities, but when working with other diverse, non-Indigenous communities as well.

To ensure the strengths of APAR methodologies can be reflected more broadly across PSM evaluation, the comprehensive multi-scale evaluation framework currently published by the first author (GYL) has been revised to strengthen the validity of the framework. The current framework seeks to understand the (i) feasibility (i.e., is PSM feasible), (ii) value (i.e., what is the value of the PSM process), (iii) change & action/impact (i.e., what changed or was actioned as a result of the PSM process), and (iv) sustainability (i.e., are the changes and actions from the PSM process sustained over time) of the PSM processes, with PAR principles embedded to support improvements of the PSM process through more equitable strategies [43]. The modified framework (Figure 1) is strengthened by embedding elements of APAR including *Indigenous epistemology* (Indigenous expertise and knowledge), *Indigenous ontology* (decolonized considerations of social and emotional wellbeing more holistically), *Indigenous axiology* (Indigenous values, laws, ethics, and procedures), and *Indigenous research methodology* (unique research and evaluation methods developed by, with, and for Indigenous peoples to promote social and emotional wellbeing) [56].

Though APAR is distinctive in that it is informed by Indigenous Standpoint Theory built by the knowledge and expertise of Indigenous social scientists in consultation with Elders and Aboriginal and Torres Strait Islander communities since the 1990′s [56], concepts of APAR complement the modified evaluation framework more broadly, and should be adopted when working with other diverse, non-Indigenous communities. Specifically, the modified framework aligns with shared knowledge and concepts of participatory evaluation, and supports a more flexible evaluation approach, as opposed to traditional rigid evaluation structures. When applied in broader PSM evaluation contexts (i.e., when working with Aboriginal and Torres Strait Islander communities as well as with other diverse, non-Indigenous communities), application of the modified evaluation framework can empower communities as co-researchers (explored via the first PSM evaluation framework criteria *feasibility*), enable the recognition of local community strengths (second PSM evaluation framework criteria *value*), ensure local priorities are nominated including nomination of evaluation points most meaningful to local communities (third PSM evaluation framework criteria *change & action/impact*), and equips local communities with the appropriate tools and resources to set future sustainable goals relevant to social and emotional wellbeing (fourth PSM evaluation framework criteria *sustainability*).

## 4. Conclusions

This paper revisits and modifies a previously published evaluation study protocol and framework to reflect APAR strategies, ensuring Aboriginal and Torres Strait Islander research and evaluation methodologies are adopted. Authors argue that these modifications further strengthen PSM evaluations more broadly. The modifications presented in this paper not only support a culturally appropriate evaluation process that amplifies opportunities to invite ongoing Aboriginal and Torres Strait Islander input, expertise, and perspectives, but it complements concepts of participatory evaluation that facilitates more empowering evaluation practices that should also be applied when working with other diverse, non-Indigenous communities. Therefore, we urge the continuous shift in literature towards prioritizing the reciprocity, strengths, and expertise that Aboriginal and Torres Strait Islander methodologies offer to broader research and evaluation practices.

## Figures and Tables

**Figure 1 ijerph-20-00053-f001:**
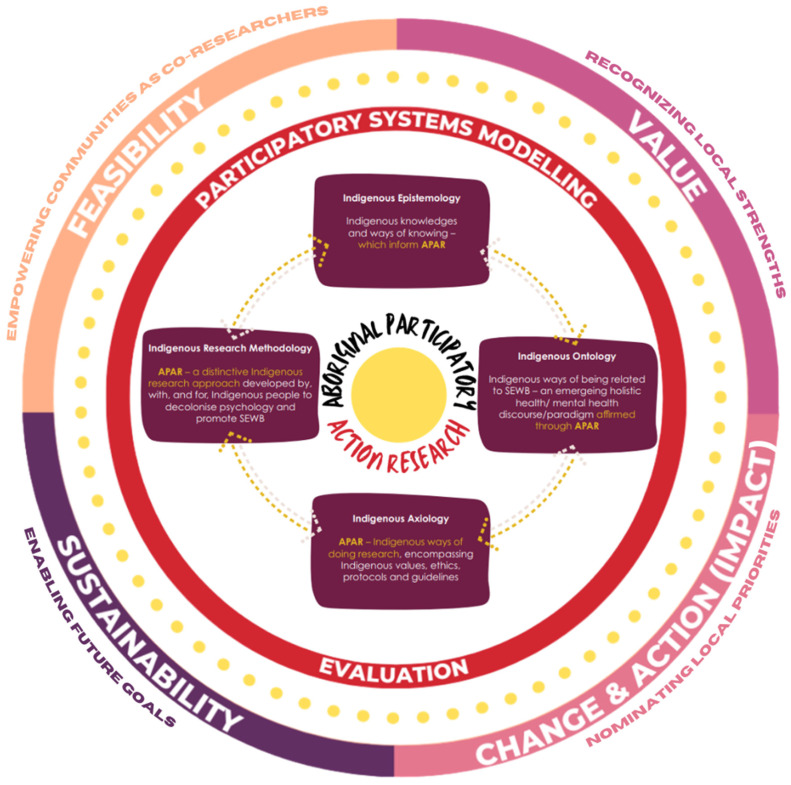
Modified PSM evaluation framework strengthened through APAR elements (i.e., Indigenous epistemology, ontology, axiology, and methodology). The inner figure depicting the elements of APAR is extracted directly from Dudgeon, Bray, Darlaston-Jones & Walker’s 2020 discussion paper on Aboriginal Participatory Action Research: An Indigenous Research Methodology Strengthening Decolonization and Social and Emotional Wellbeing [56]. The presented modified framework has shared concepts of participatory evaluation which has broader applications including PSM evaluation with Aboriginal and Torres Strait Islander peoples and with other diverse, non-Indigenous communities.

## Data Availability

Not applicable.

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
