# Peer review of "Partnering with Aboriginal and Torres Strait Islander Peoples: An Evaluation Study Protocol to Strengthen a Comprehensive Multi-Scale Evaluation Framework for Participatory Systems Modelling through Indigenous Paradigms and Methodologies"

_ijerph, 2022, doi:10.3390/ijerph20010053_

Round 1

Reviewer 1 Report

Thank you for the opportunity to read and review your paper. I very much enjoyed reading it and find it to be expertly written and in it’s current form consider it to be ready for publication. It is good that you are publishing this in the IJERPH, as I believe it fits the scope of the journal well and this open access international format should/will maximise access to it.

Some feedback written as I read and reflected on your paper:

Powerful Acknowledgements in the opening of the introduction. The statement re: the mix of Indigenous and non-Indigenous co-authors reads as respectful and appropriate and speaks to the power of working at the Cultural Interface. Thank you.

Was shocked and saddened at the suicide rates for Aboriginal and Torres Strait Islander adolescents your report, despite knowing the national trends generally.  The introduction stats are a dark reflection of the reality of how little the recent shifts from the white deficit problematisation lens on Indigenous health and culture, to strengths based and positive rhetoric, have still not addressed/reduced institutional and individual racism and the impacts of colonisation and inertia to change. 

The ‘Right care, first time, where you live’ program reads as a highly effective and much needed research in the context of enhancing the SEWB of young Aboriginal and Torres Strait Islander peoples. It’s bottom-up approach, beginning with local community empowerment aligns with Indigenous research paradigms, and trialling it across 8 (reflecting the 8 ways approach?) diverse communities from urban to rural speaks to cultural  diversity, thus is likely to achieve the aim of empowerment and delivery to local community needs.

The phrase you have cited from the US  “quiet movement to make government fail  less often” speaks volumes here and certainly enriches your description of the Program.  Totally support your argument that Indigenous programs remain inappropriate and inadequately planned to enable Indigenous knowledges and methodologies to be the main informants and contributors in the evaluation field, and that the majority of government and mainstream  institutional driven evaluations and reports  further embed the voices of mainstream approaches that continue the harms of colonisation.

The study design reads well, and being informed by Indigenous Standpoint Theory is a strength.  The issues you raise around Indigenous data sovereignty are timely and in urgent need of acknowledgement and increased respectful discussion by mainstream academia. Data analysis, security and your Ethics statements read as clear and appropriate.

Figure 1 is very effective. Easy to understand and clear. It adds much to the paper. It resonates strongly with and mirrors the original diagram drawn by  Tristan Schulz Gee, Dudgeon, Schultz, Hart and Kelly, (2013)

Your discussion is clear and concise and does not get bogged down in theory and policy, meaning it can be easily read and understood by the reader, with clear take away messages stated. It is to be hoped this paper and your work will be widely accessed by both Indigenous and non-Indigenous policymakers and researchers and beyond. Your use of participatory systems modelling has potential to be a game changer in urging the current inertia of institutions and governments to move beyond mere box-ticking attitudes, to a genuine movement to fund decolonising of evaluations led by community with Indigenous governance and design. Reflexive practice by institutions and systems remains a major challenge, however I believe this work may well be one of the building blacks of changing that.

Your paper makes a welcome contribution to the current knowledge and literature in the area of SEWB and the advancement of Indigenous knowledges and research paradigms.

The only suggestion I can make  is that (given your use of mixed methods in your study) you consider a mention of or citation to the work of Maggie Walter & Chris Anderson and their book Indigenous Statistics, (2013), the first book ever published on Indigenous quantitative methodologies.  This book is a powerful example and evidence of the strengths and high value of Indigenous knowledges in quantitative research.  However this is a suggestion only, and I still recommend this paper is ready to go in it’s current form.

Author Response

Many thanks for your thorough review, and kind feedback. Thank you also for your recommended citation – what an interesting read! Chapters 3 and 4 of Walter & Anderson’s book on the paradigm of quantitative Indigenous methodologies and its application in practice particularly enhanced our understanding. We have cited Indigenous Statistics in our 2. Materials and Methods section, under 2.4 Data collection process. These changes are tracked in our word document to support a more enjoyable reading experience on line 388 and on lines 390-391.

Reviewer 2 Report

The paper is dealing with a very important topic and has much potential. However in its present form it reads as a project proposal rather than report on results achieved. It talks about what should be done, what will be done, but contains almost no information about what has been done. We all agree that it is extremely important to engage with indigenous people and involve them in the research, and participatory modeling is exactly the kind of framework that can allow such co-learning to take place. But what has already been done? What was achieved? Do we have any real case studies to report? What is the added value of the paper?

Author Response

Thank you for your feedback. It is correct that the manuscript is a study protocol, not an original research paper. We are excited to disseminate our findings over the years to come, working closely with local Aboriginal and Torres Strait Islander leaders as well as other Indigenous academics and experts (who are authors on this paper) on the best ways to share our learnings. Publishing on our protocol first before reporting on our findings is critical to increase the transparency and validity of our approach, as well as to also allow others to learn and build on our approach.

Reviewer 3 Report

Thank you for the opportunity to review this paper. It is really great to see these ideas and approaches being considered and adopted in the early stage of such a large and influential program – and for these to be disseminated so others can adopt similar approaches. I have made a small number of comments for consideration, but overall well written and thoroughly referenced piece, with very good Aboriginal and Torres Strait Islander oversight and guidance.

1.     Introduction

It might not be strictly true to say that Aboriginal and Torres Strait Islander peoples is ‘preferred terminology in Australia’ – or it may be preferred terminology of institutions but not necessarily of individuals who feel strongly about how they wish to be identified and do not want to be grouped in a collective term. Because this is an international journal, and preferences around language will evolve over time (and may shift in Australia to more common usage of First Nations) it might be better to say “in this paper we use the term Aboriginal and Torres Strait Islander peoples in reference to the First Peoples of Australia”. Perhaps adding a note respectfully acknowledging the diversity and heterogeneity across the different communities.

1.1  SEWB

This section describes poor SEWB of young Aboriginal and Torres Strait Islander people in detail and the many contributing factors. It would be useful to also include some information on what good SEWB looks like and where possible provide some evidence and statistics of what we are aiming for and the benefits for a young person who has good SEWB and the conditions needed for that. The abstract makes reference to “protective and healing factors of social and emotional wellbeing”. Some description of these, the cultural resilience of Aboriginal and Torres Strait Islander people in the face of ongoing colonisation and the benefit (and need) for connections with community, culture, Country would be good. At the moment this sectioned is framed primarily around a vulnerable population and that framing itself can contribute to poorer mental health outcomes. See also Lowitja publications:

Deficit Discourse and Indigenous Health: How narrative framings of Aboriginal and Torres Strait Islander people are reproduced in policy

https://www.lowitja.org.au/page/services/resources/Cultural-and-social-determinants/racism/Deficit-Discourse-and-Indigenous-Health

Deficit Discourse and Strengths-based Approaches: Changing the Narrative of Aboriginal and Torres Strait Islander Health and Wellbeing

https://www.lowitja.org.au/page/services/resources/Cultural-and-social-determinants/racism/deficit-discourse-strengths-based

1.2  Participatory methods

I wonder if there is anything that could be said about:

1) the importance of system modelling where it is the system that has failed Aboriginal and Torres Strait Islander people and that big structural changes are needed. This approach assists with identifying where and how that is possible. Could also be useful to explicitly mention racism as a systemic issue somewhere, in the same way this was acknowledged upfront in the way that Indigenous peoples have been excluded from academia.  

2) would it be true to say systems modelling fits with Indigenous perspectives of seeing the whole system and not focusing on individuals/individual failings?

Lines 169-172 – I found this sentence a bit hard to follow if it could be tweaked slightly “Importantly, the PSM process focuses on working with local community stakeholders as co-researchers, empowering communities to prioritize what should be incorporated into the system dynamics models, such as which programs and services should be included within the model for simulation.”

Perhaps because at this stage in the protocol the process hasn’t been described in detail it’s not clear how ‘prioritize what should be incorporated into the system dynamics models’ is different from ‘which programs and services should be included within the model for simulation’

217 – Australian Government or Australia’s Commonwealth Government?

2.     Materials and Methods

The partnership with Aboriginal and Torres Strait Islander community members is great to see. A little more could be said about how this particular project is a community priority. Some of this is hinted at in lines 308-310 but could be clearer what processes are in place here.

Perhaps review and align this and future publications against the CONSIDER statement to ensure the inclusion of Indigenous peoples is reflected, including governance arrangements (noting later in the piece this states there will be local protocols for each site - be good to be explicit about the overarching governance)

For reference: Consolidated criteria for strengthening reporting of health research involving Indigenous peoples: the CONSIDER statement https://bmcmedresmethodol.biomedcentral.com/articles/10.1186/s12874-019-0815-8

2.3 IDS

Really pleased to see some specifics of IDS outlined here

2.4 Data collection processes

The description of different Indigenous research methods is useful as is the acknowledgement that these may not be appropriate for a non-Indigenous researcher. Is it possible to ensure that an Aboriginal and/or Torres Strait Islander researcher from each site is co-leading the evaluation in that setting?

Author Response

 Thank you for your kind words of encouragement. We are committed to ensure that all aspects of our program remain transparent so others can learn and build on our approach, and we appreciate your support to publish this work. Thank you also for spending the time to provide detailed feedback. We have incorporated your suggestions as follows (marked in orange font):

  1. Introduction: We have changed our acknowledgement to align with your suggestions as follows:

We recognize that Aboriginal and Torres Strait Islander peoples is utilized throughout the paper to reference to the First Peoples of Australia the preferred terminology in Australia. However, First Nations and Indigenous are also utilized throughout this paper when reflecting an international context. 

1.1 SEWB: Thank you for your important suggestion. We have expanded this section to reflect a more strengths-based narrative on SEWB. These changes are tracked in our manuscript to support a more enjoyable reading experience on lines 92-95.

1.2 Participatory methods: Thank you for your suggestions. We agree with all of your points, and as such:

1) We have made more explicit that it is the system that has failed Aboriginal and Torres Strait Islander people – specifically noting cultural genocide, racism, and colonisation as key factors – and that significant structural changes are required to enable positive change. We also then tie this into how participatory systems modelling can support systems changes. This is noted in line 101, and in lines 139-141.

2) Yes, it is true that systems modelling fits with Indigenous perspectives of seeing the whole system and not focusing on individuals/individual failings. Thank you for this fantastic suggestion! We have included this in as a tracked change in our updated manuscript on lines 151-152.

3) Regarding lines 169-172: Thank you for pointing out that it was difficult to follow. We have modified this section to align with the previous sentences. Specifically, the previous sentences (on lines 175-178) describe how the models can simulate hypothetical ‘what-if’ scenarios of likely health, social, and economic impacts of individual (e.g., ACHHS) or a combination of programs and services (e.g. ACCHS combined with community-based acute care services). Lines 180-184 have been modified to tie in with how the PSM process will focus on working with community stakeholders as co-researchers to prioritise what should be reflected in the models, such as which programs and services should be included within the model to simulate various ‘what-if’ health, social, and economics scenarios.

5) Thank you for your suggestion regarding line 217. We have accepted your suggestion and now describe the Australian Commonwealth as Australia’s Commonwealth Government (line 229).

  1. Material and Methods: Thank you for your important suggestions.

We have expanded lines 322-324 to more explicitly mention that partnerships with Aboriginal and Torres Strait Islander community members as co-researchers ensures that our research Program reflects the priorities of local Indigenous communities rather than that of the Program research team.

Your point on outlining more specifically the processes put in place to enable genuine partnerships with Aboriginal and Torres Strait Islander community members is such an important one. Thank you also for recommending Huria et al.’s fascinating paper on the CONSIDER statement. Our specific processes are explicitly stated throughout our 2. Materials and Methods section – specifically, under the various sub-sections. Upon reading Huria et al.’s paper, we received affirmation when recognising our processes align with the CONSIDER checklist domains including research governance, prioritisation, partnerships, methodologies and methods, participation, capacity, analysis and interpretation, and dissemination. With that said, we have expanded our 2. Materials and Methods section more broadly to reference this critical work throughout (reference 60). Specifically:

  • Line 322
  • Line 324
  • Line 327
  • Line 342
  • Line 423

2.3 IDS: Thank you for your kind words. We are honoured that you feel IDS is appropriately outlined in our paper.

2.4 Data collection processes: Thank you for your important suggestion. We have expanded this section on line 397 by stating that the Program team will work under guidance from local Aboriginal and Torres Strait Islander leaders as well as other Indigenous academics and experts (who are authors on this paper), on which techniques are appropriate to implement by whom (e.g., co-led by Aboriginal and Torres Strait Islander researcher from the site).  

Round 2

Reviewer 2 Report

My  comments have not been addressed by the authors, so I see no reason to change my  opinion about the paper. The numerous comments on the manuscript have not even been considered. 

Author Response

Thank you for your feedback, and we apologise that you feel that your comments have not been addressed.

Please see enclosed our response to the reviewer, which provides a point-by-point response to comments. Please also refer to our enclosed manuscript which addresses the reviewer's feedback further. 
